# Uranium Concentrations in Private Wells of Potable Groundwater, Korea

**DOI:** 10.3390/toxics10090543

**Published:** 2022-09-18

**Authors:** Woo-Chun Lee, Sang-Woo Lee, Ji-Hoon Jeon, Jong-Hwan Lee, Do-Hwan Jeong, Moon-Su Kim, Hyun-Koo Kim, Soon-Oh Kim

**Affiliations:** 1HS Environmental Technology Research Center, Hosung Inc., Jinju 52818, Korea; 2Department of Geology and Research Institute of Natural Science (RINS), Gyeongsang National University (GNU), Jinju 52828, Korea; 3Soil & Groundwater Research Division, Environmental Infrastructure Research Department, National Institute of Environmental Research (NIER), Incheon 22689, Korea

**Keywords:** natural radioactive elements, uranium, groundwater, private wells, potable use, geology

## Abstract

Uranium (U) is one of the typical naturally occurring radioactive elements enriched in groundwater through geological mechanisms, thereby bringing about adverse effects on human health. For this reason, some countries and the World Health Organization (WHO) regulate U with drinking water standards and monitor its status in groundwater. In Korea, there have been continuous investigations to monitor and manage U in groundwater, but they have targeted only public groundwater wells. However, the features of private wells differ from public ones, particularly in regard to the well’s depth and diameter, affecting the U distribution in private wells. This study was initiated to investigate U concentrations in private groundwater wells for potable use, and the significant factors controlling them were also elucidated through statistical methods. The results obtained from the analyses of 7036 groundwater samples from private wells showed that the highest, average, and median values of U concentrations were 1450, 0.4, and 4.0 μg/L, respectively, and 2.1% of the wells had U concentrations exceeding the Korean and WHO standard (30 μg/L). In addition, the U concentrations were highest in areas of the Jurassic granite, followed by Quaternary alluvium and Precambrian metamorphic rocks. A more detailed investigation of the relationship between U concentration and geology revealed that the Jurassic porphyritic granite, mainly composed of Daebo granite, showed the highest U contents, which indicated that U might originate from uraninite (UO_2_) and coffinite (USiO_4_). Consequently, significant caution should be exercised when using the groundwater in these geological areas for potable use. The results of this study might be applied to establish relevant management plans to protect human health from the detrimental effect of U in groundwater.

## 1. Introduction

Uranium (U) primarily originates from geological events and is one of the typical natural radioactive elements that occur in groundwater. U occurs as mineral phases, such as uraninite and monazite, and the dissolution of such U-bearing minerals dominantly influences groundwater quality in terms of U contamination. It is well-known that U is mostly enriched in deep hydrothermal deposits, granitic pegmatites, and carbonate sedimentary rocks [1]. In addition to its natural origins, U contamination in groundwater is also caused by various anthropogenic sources, such as nuclear power plants, phosphorus fertilizers, and mining activities [2]. Even though U concentrations are relatively low in groundwater, chronic adverse effects, such as kidney toxicity, can be caused by the long-term intake of such groundwater. Uranium toxicity is manifested in two manners, chemical toxicity and radiotoxicity, and the former is known to be six times larger than the latter because of uranium’s long half-life (i.e., 45 billion years for ^238^U) [3]. For these reasons, the regulations of the World Health Organization (WHO) and several advanced countries state that the U level of drinking water should not exceed 30 µg/L due to chemical aspects of uranium toxicity [4,5]. In addition, regarding its radiological aspects, the guidance levels of U radionuclides are addressed as 10 and 1 Bq/L for ^238^U and ^234^U, respectively [5]. Moreover, Korea set the U standard (30 µg/L) for drinking water in 2019.

The level of natural radioactive U in groundwater has been continuously monitored worldwide over the last 50 years. Levels in drinking water are generally less than 1 μg/L, although concentrations as high as 700 μg/L have been measured in private supplies [5]. In aaddition, Vengosh et al. (2022) reported that the U concentrations in groundwater ranged from 0.0 to 10.0 µg/L based on an investigation of 46 countries [6]. Because most U contaminations in groundwater are related to geological origins, geology is crucial in controlling them. Among various geologies, granite showed relatively higher U concentrations. In addition to areas of granite, the groundwater quality in terms of U contamination is significantly influenced by the mineral compositions of granite, the hydrochemical properties of groundwater, and the features of U-bearing minerals (occurrence phases, speciations, and solubility) [7,8]. It has been reported that the level of U concentrations was comparatively higher in granite areas in Korea [9,10], especially in two-mica granites where U is highly enriched, which are mainly distributed in the central district [11]. According to Jeong et al. (2011), groundwater contamination by U was caused by the dissolution of minor constituents accompanied by biotite [12]. In particular, Shin et al. (2016) reported that 160 (3.9%) out of 4140 groundwater wells showed U concentrations exceeding the drinking water standard (30 µg/L) in Korea [13]. Furthermore, Vensogh et al. (2022) reported that the U concentrations in groundwater are likely higher in Korea than in other countries [6]. Among the countries investigating more than one hundred groundwater wells, India showed the highest average and median values of 20.75 and 5.21 µg/L, respectively, and relevant countermeasures should be considered [6,14,15]. The investigations conducted in 1998 revealed severe U contamination of groundwater in Daejeon city, the fifth biggest city in Korea, which led to the launching of comprehensive and systematic surveys. The Ministry of Environment of Korea (KMOE) conducted the first phase survey that lasted for four years (from 1999 to 2002), and the status of U contaminations in groundwater was investigated, focusing on the groundwater used as a source of drinking water for small-scale public supply facilities in a couple of local areas [16]. Subsequently, the KMOE initiated the second phase survey that lasted for 12 years (2007–2018), and it was extended to nationwide public drinking-water supply facilities that used groundwater sources [17]. At present, the third phase survey on public facilities is ongoing. However, many people rely on private wells for potable groundwater sources rather than public ones. Nevertheless, in the surveys undertaken so far, only public wells have been targeted. The characteristics of private wells differ significantly from those of public ones, so the tendency of U contamination might also differ. This lack of investigations into private wells for potable groundwater led to this study.

This study was launched as the first attempt to investigate the current status of U contamination in private wells for potable groundwater in Korea. The field survey was undertaken using field measurements of the representative hydrochemical properties. The groundwater samples were taken and analyzed in terms of field constituents (pH, temperature, redox potential, electrical conductivity, and dissolved oxygen content) and major and minor constituents to relate elevated U concentrations to geochemical factors. Furthermore, U concentrations were measured for each sample and categorized into groups based on their level to elucidate the relationship between them and the geology of the site using normalized correlation analyses. Finally, the geologies where U concentrations appeared to be relatively higher were classified as requiring more detail to scrutinize the origin of the high U concentration.

## 2. Materials and Methods

### 2.1. Sampling and Geology of the Study Area

The total number of target wells was 7036, and their locations are given in Figure 1. A field survey was conducted in 2021 with geological maps that had scales of 250,000:1 and 50,000:1. Based on the geology presented by the 250,000:1 map, the geology and geological period of the well-sitting sites were largely classified into 11 groups (Table 1 and Table 2). Subsequently, the 50,000:1 geological map was used to identify detailed information on the geology of the U-enriched sites, such as the specific types of rocks and their major mineral constituents.

### 2.2. Field Measurements and Chemical Analyses

A five- or ten-minute purging of the groundwater wells was conducted before sampling to prevent the groundwater from being contaminated and altered. The primary hydrochemical properties of the groundwater were measured in the field, and the major and minor constituents, were measured in the laboratory. The pH, electrical conductivity (EC), standard redox potential (Eh), and dissolved oxygen content (DO) of the groundwater samples were measured in the field using a portable multi-parameter meter (ProQuatro, YSI, Yellow Springs, OH, USA). Furthermore, the bicarbonate (HCO_3_^−^) concentrations were measured in the field using the standard acid-titration method [18].

The groundwater samples were filte”ed t’rough 0.45-μm membranes in the field and then placed in 125-mL PE bottles. The samples for the analyses of cations and U were acidified using concentrated nitric acid to prevent their adsorption and coprecipitation during transportation. The major cations (Na, K, Mg, and Ca) and Si were measured using an inductively coupled plasma-optical emission spectrometer (ICP-OES, PerkinElmer, BRO Avio 500, Shelton, CT, USA). The anionic species (F, Cl, SO_4_, and NO_3_) were analyzed by ion chromatography (IC, Thermo, ICS-1100 aquion, Sunnyvale, CA, USA). The hydrochemical facies of each groundwater sample were determined using Aquachem v.4.0 (Waterloo Hydrogeologic, Inc., Waterloo, ON, Canada), and the results are presented in Table 3. The U concentrations were measured using an inductively coupled plasma-mass spectrometer (ICP-MS, Agilent, 7700, Sunnyvale, CA, USA), and Table 4 shows the results.

### 2.3. Correlation Analyses through PCA

The relationship between the hydrochemical parameters and U concentrations was evaluated using correlation analyses normalized by principal component analysis (PCA) instead of conventional correlation analyses. The PCA-normalizing correlation method was used in this study to improve the reliability and significance level [19,20,21]. The normalized correlation analyses were conducted using the SigmaStat 4.0 program of the Sigmaplot 14.0 package (Systat Software Inc, San Jose, CA, USA). The normalization of the correlation matrix was undertaken following the method explained by Jolliffe and Cadima (2016) [19]. In short, the normalized method could correct and improve the results of the conventional one by applying the coefficients of the linear combination obtained from the PCA to the correlation matrix.

## 3. Results

### 3.1. Hydrochemical Properties of Groundwater

#### 3.1.1. The Results of Field Measurements

The results of the field measurements of the 7036 private wells are given in Table 1. The average well depth was estimated to be 69.4 m, which was comparatively shallower than that (113 m) of the previous survey on groundwater wells used by small-scale public supply facilities [13]. The average pH values of the Paleozoic sedimentary rocks (Ps) and Mesozoic sedimentary rocks (Ms) were 7.4 and 7.3, respectively, and they were the highest among the results. The other geologies showed average pH values of 6.6–7.1. The higher pH in the sedimentary rocks is attributed to increased bicarbonate concentrations due to the dissolution of carbonate rocks [22]. The average Eh (317 mV) appeared to be highest in the Precambrian sedimentary rocks (PCs), and the samples of the unknown era showed the lowest average Eh value (153 mV). The average Eh values ranged from 195 to 344 μS/cm in most geologies, and the Ps showed the highest value (351 μS/cm). Moreover, the study by Zhang et al. (2020) suggested that the EC was relatively higher in the area of sedimentary rocks compared with other rocks [23]. The DO values ranged from 3.4 to 5.6 mg/L, which were likely to be irrelevant to geology. In summary, the field measurement parameters were generally higher in the sedimentary rocks, which might be attributed to the significant solubility of the minerals included in them [22,23].

#### 3.1.2. The Concentrations of Major and Minor Constituents

The average concentrations of major cations were increased in the order of Ca > Na > Si > Mg > K (Table 2), and these results were consistent with the results reported by Lee et al. (2020) [24]. Regarding the average concentrations of major anions, bicarbonate showed the highest concentration, and those of the other constituents were NO_3_ > Cl > SO_4_ (Table 2).

The relationship between the geology and major and minor Ion concentrations elements was examined hereafter. The Na concentrations were lowest in the Mesozoic igneous rocks (Mi) at 3.3 mg/L, and the other geologies showed the range of 6.1–18.3 mg/L. The lower value of Na concentration in igneous rocks might be caused by the lower solubilities of albite and plagioclase [25,26]. However, irrespective of geology, the K, Mg, and Ca concentrations showed somewhat narrower ranges of 0.9–1.8 mg/L, 5.0–11.1 mg/L, and 21.4–45.6 mg/L, respectively. In the case of Si, the PCs showed the highest average value (19.5 mg/L), and the Si concentrations ranged from 5.3 to 15.3 mg/L in the other geologies. The solubility of quartz, amorphous silica, and other Si-bearing minerals increased with increasing pH but decreased in neutral and acidic conditions. Therefore, the highest Si concentrations in ”he s’Iimentary rocks could be attributed to the higher pH values of those rocks [24].

In terms of anionic species, the average concentrations of bicarbonate were measured to be 75.7–154.0 mg/L, and its highest concentration appeared in the Mesozoic igneous rocks (Mi), resulting from the dissolution of carbonate minerals, as mentioned above. In the case of sulfate concentrations, the Jurassic granites (Jgr) and the PCs rendered comparatively higher concentrations of 27.2 and 23.0 mg/L, respectively, and ranged from 11.3 to 16.0 mg/L in the other geologies. Gypsum and sulfide minerals, such as pyrite and chalcopyrite, are the primary sources of sulfate ions. In particular, the solubility of gypsum is overwhelmingly higher than the others, and the SO_4_ concentrations seem to be dominantly controlled by the content of gypsum. These coupled effects elevate SO_4_ concentrations in the sedimentary rocks [23]. The average nitrate (NO_3_) concentrations were the highest in the PCs at 30.3 mg/L, and the average values ranged from 11.1 to 28.9 mg/in the other types of rocks. It is well-known that the primary source of nitrate originates from agricultural activities, such as fertilizers, and the effects of geology on its concentrations are likely to be insignificant [27]. The average concentrations of fluorine (F) and chlorine (Cl) were lower than 0.5 mg/L and 6.5–23.8 mg/L, respectively, and they seemed irrelevant to geology. Furthermore, the results of F and Cl indicated that the groundwater in the study area could be classified as fresh water without any effects of seawater intrusion.

#### 3.1.3. Hydrochemical Facies of Groundwater

Overall, Ca–HCO_3_ types were predominant, and their proportion was 74.3% and followed by Na–HCO_3_ (12.8%) and Ca–NO_3_ (7.0%) (Table 3). In terms of the relationship between hydrochemical facies and geology, regardless of geology, the Ca–HCO_3_ types of groundwater were general, indicating that most of the groundwater belonged to typical fresh water, as mentioned above. These results were congruent with the results of Shin et al. (2016) and Lee et al. (2020), in which the Ca–HCO_3_ type of groundwater was dominant in the study area [13,22]. In the case of sedimentary rocks, Ps and Ms showed relatively higher proportions of Ca–HCO_3_-type groundwater, whereas the PCs presented a lower proportion (Table 3). It might be caused by the lack of supplying carbonate minerals due to the termination of their weathering [28]. The Na–HCO_3_ types occupying the second highest portion were relatively higher in the Cretaceous granite (Kgr), resulting from the dissolution of silicate minerals of igneous rocks [29].

### 3.2. Uranium Distribution According to Geology

The analytical results for the 7036 groundwater samples indicated that the U concentrations ranged from the concentration lower than the detection limit to 1450 µg/L, with the average and median values of 0.4 µg/L and 4.0 µg/L, respectively (Table 4). Shin et al. (2016) measured the U concentrations of 4140 groundwater wells and reported that the U concentrations were approximately two times higher than in this study (0.7 µg/L and 8.0 µg/L for the average and median values, respectively) [13]. Shin et al. (2016) targeted public groundwater wells for drinking water and found that they were likely to be deeper than those of this study of private wells, probably resulting in the elevated U concentrations. Moreover, the results of this study indicated that the U concentrations increased with increasing well depth, as shown in Figure 2. The number of wells with U concentrations exceeding the Korean and WHO criterion for drinking water was 148, and the excess rate was computed to be 2.1%, which was lower than the results of Shin et al. (2016), which showed an excess rate of 3.9% [13]. Jgr was the geology with the highest excess rate, and 107 wells exceeded the standard, resulting in a 3.4% excess rate. Followed by Jgr, the excess rates were higher in the order of Qa (1.5%) > PCm and Kgr (1.3%) > Og (1.1%) (Table 4). Hwang et al. (2016) also reported that higher U concentrations were measured in the geologies of the Jgr and Pcm [30], and the results seemed to be consistent with ours.

Among all the target wells (7036), 4707 (66.9%) wells showed U concentrations below 1.0 µg/L, and they were neglected in this study in order to focus on more effective data (Table 5). This premise might be supported by Vengosh et al. (2022) [6]. They investigated U concentrations in groundwater across 46 countries and reported that the median value of the U concentrations was 0.89 µg/L. Moreover, they proposed 1.0 µg/L as the background U concentration and found that if the U concentrations are higher than 1.0 µg/L, then there is strong evidence to support a higher probability of U contamination. Using the results higher than 1.0 µg/L, the U concentrations were categorized into five groups depending on their levels. Table 5 and Figure 3 show these five groups of U concentrations for three geologies having the highest excess rates. In the Jgr geology, the U concentrations increased with increasing sample numbers, but the opposite tendency was observed in the case of PCm and Qa (Figure 3), indicating that U was dominantly distributed as the higher level of concentrations in the Jgr, but as the lower level in the PCm and Qa. In other words, the groundwater in the Jgr area seemed more susceptible to U contamination.

### 3.3. U enrichment Mechanism

For the three geologies showing the highest excess rates, the median U concentrations of the five groups are presented with all the other parameters in Table 6. The median values were used rather than the average ones because the data showed skewed distribution and wide variations [31]. In the Jgr, the U concentrations were increased with increasing EC, Ca, and HCO_3_ and decreasing Eh, and the other parameters did not show any tendency with U concentrations. In the case of PCm, the negative correlation between U and Cl/NO_3_ was notable, and there was no other correlation. Finally, any remarkable correlation between U and the parameters was not observed in the Qa. Therefore, the relationship between U and the other parameters was much more remarkable in the Jgr than in the other two geologies. To more clearly scrutinize their relationships, the correlation analyses normalized by PCA were conducted, and the results are given in Table 7. The results of the normalized correlation analyses indicated none of the distinct correlations between them.

Jgr proved to be the most crucial geology related to the elevated U concentrations among eleven geologies of the study area, and its specific rock types were investigated in more detail using the geological map of a larger scale (50,000:1), focusing on the E group. Figure 4 shows the distribution of U concentrations according to the detailed geologies of the Jgr. Among the E groups of the Jgr, the Jurassic biotite granite (Jbgr) showed the most significant sample number (73), followed by the Jurassic two-mica granite (Jtgr) > Jurassic porphyritic granite (Jpgr) > Jurassic hornblende–biotite granite (Jhgr) > Jurassic gneissoid hornblende biotite granite (Jggr). However, the average U concentration of the Jpgr appeared overwhelmingly higher than the other geologies. The higher level of U concentrations in these granitic rocks is attributed to the containment of U-bearing minerals, such as uraninite, during the formation of parental granite or the high content of U in biotite, the primary mineral in granite [32,33]. The Daebo granite belonging to the Jpgr showed a comparatively higher U concentration, and this result corresponds with those of Hwang et al. (2018) [34]. In addition, they revealed that the primary sources of U were uraninite (UO_2_) and coffinite (USiO_4_), included in the Daebo granite. Overall, the Daebo granite seemed to be a predominant source of U in Korea. During the differentiation of the Jurassic granite, the order of crystallization is granodiorite, biotite granite, and two-mica granite, and U is enriched in these granitic rocks because U is one of the typical large-ion lithophile or incompatible elements excluded from the early-stage crystallization of magma and enriched in the rocks crystalized in the last stage. Kim et al. (2022) reported that natural radioactive elements, including U, were slowly leached out from the edge of the biotite of granite for an extended period, resulting in groundwater contamination [35]. The U concentrations of the two-mica granite (Jtgr) were much higher than the biotite granite (Jbgr) (Figure 4), which is attributed to the crystallization order (i.e., the Jtgr is crystallized at the last stage of magma differentiation).

No significant relationships between the U concentrations and hydrochemical parameters were observed, based on the results of the normalized correlation analyses for the three geologies showing relatively higher excess rates (Table 7). Therefore, focusing on the primary geology, the Jgr, showing the highest excess rates, the correlation analyses were conducted with the dataset of U concentrations belonging to the E group, and the results are given in Figure 5. In the results, three sub-groups of the Jgr, such as the Jbgr, Jtgr, and Jpgr were highlighted. In the case of the Jpgr, the correlation factors of U with major cations and anions were estimated to be negative, and its absolute values were larger than 0.4, which differed from those of the Jbgr and Jtgr. The dissolution of uraninite was the likely cause due to water rock interaction, and only the U concentrations were increased without changing the concentration of major cations and anions. In particular, all three rock types showed excellent correlations between U and pH, which was probably due to the increase in the concentration of UO_2_(CO_3_)_3_^4−^ with increasing pH [13] (Shin et al., 2016). The bicarbonate (HCO_3_) could react with UO_2_ in groundwater, and its concentration was decreased. Therefore, the negative correlation with HCO_3_ was observed in the Jpgr, despite the positive correlation with pH. Compared with Jpgr, Jbgr and Jtgr showed an insignificant correlation between U and the other parameters because the U concentrations seemed to increase as a result of the leaching of U from the edge of biotite and mica rather than the direct dissolution of these two minerals.

## 4. Conclusions

The quality of groundwater can deteriorate due to naturally occurring radioactive substances, such as U, and systematic investigations are required to guarantee its safety, especially for drinking use. For this reason, the Ministry of Environment of Korea (KMOE) first launched a continuous related project in 1998. The previous reports revealed that the status of uranium contaminations in groundwater could not be left unresolved any longer. Based on the previous survey results, the KMOE established the standard (30 μg/L) for uranium levels in drinking water in 2019 in an effort to solve the problem. However, the investigations have only targeted groundwater wells for public supply, even though many people rely on private wells for potable groundwater sources. This situation led to the initiation of this study.

In this study, 7036 private wells used for potable groundwater were investigated, and the results showed that the maximum, median, and average concentrations of uranium were 1450 µg/L, 4.0 µg/L, and 0.4 µg/L, respectively. Overall, 148 wells showed uranium concentrations exceeding the drinking water standard, resulting in an excess rate of 2.1%. The geology showing the highest excess rate was Jurassic granites, followed by Precambrian metamorphic rocks and Quaternary alluvia. Among various types of Jurassic granites, Jurassic biotite granite, Jurassic two-mica granite, and Jurassic porphyritic granite were intimately related to the higher levels of uranium concentrations. In particular, the average uranium concentration of the Jurassic porphyritic granite appeared overwhelmingly higher than the other geologies, and the Daebo granite belonging to this type of granite showed a comparatively higher uranium concentration. In addition, the previous study revealed that the primary sources of U were uraninite (UO_2_) and coffinite (USiO_4_), included in the Daebo granite. Overall, Daebo granite seems to be a predominant source of U in Korea.

A comprehensive review of the removal of uranium by water treatment processes has been undertaken, and the results reported that the most effective treatment technologies specific to uranium are coagulation, precipitation/softening, ion exchange, and reverse osmosis. However, those technologies are not likely feasible at private level, and sitting groundwater wells in safe areas is likely to be the most practical solution at present. Consequently, the geology of the corresponding areas for developing groundwater wells, particularly for potable use, should be considered. In this respect, the results of this study will serve as fundamental and valuable data that could be applied to establish relevant management plans to protect human health from the detrimental effect of uranium in groundwater.

## Figures and Tables

**Figure 1 toxics-10-00543-f001:**
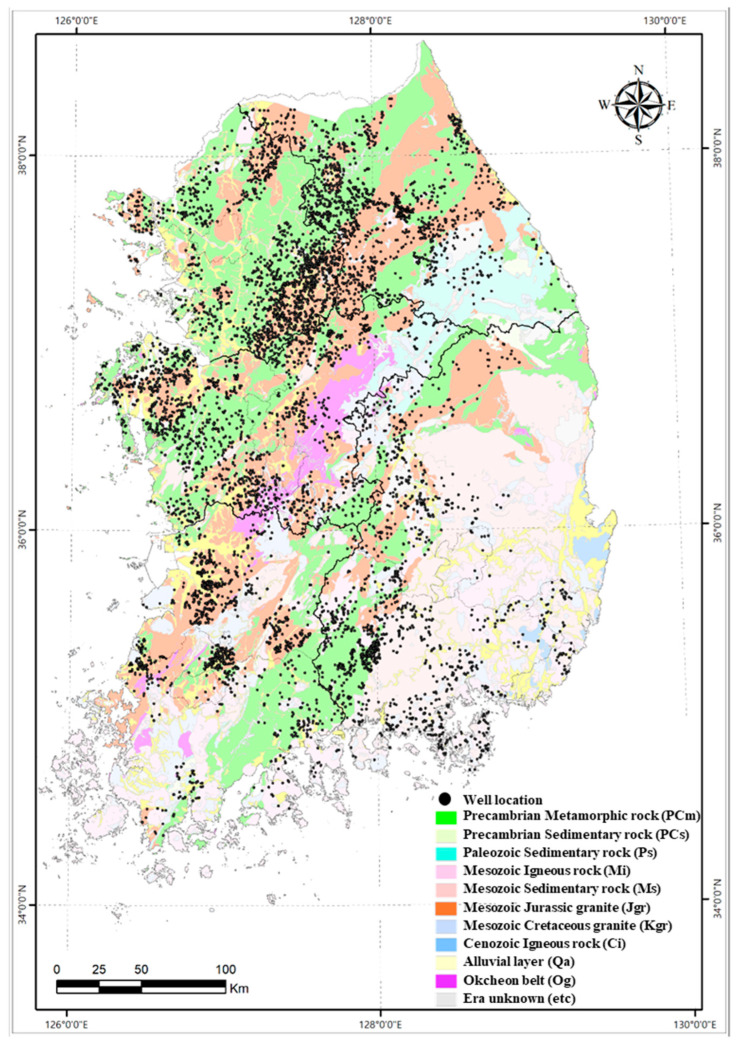
The locations of each groundwater well shown on the geological map having the scales of 250,000:1.

**Figure 2 toxics-10-00543-f002:**
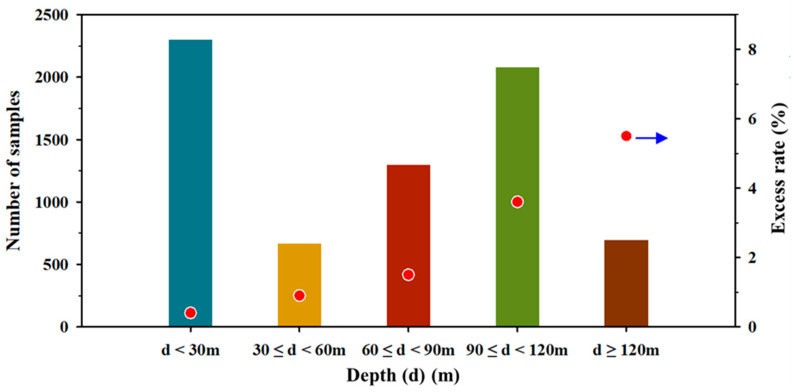
The number of samples and the excess rate of uranium according to groundwater depth. Data from Table 2 and Table 4. The blue arrow designates the value of the *y*-axis on the right side.

**Figure 3 toxics-10-00543-f003:**
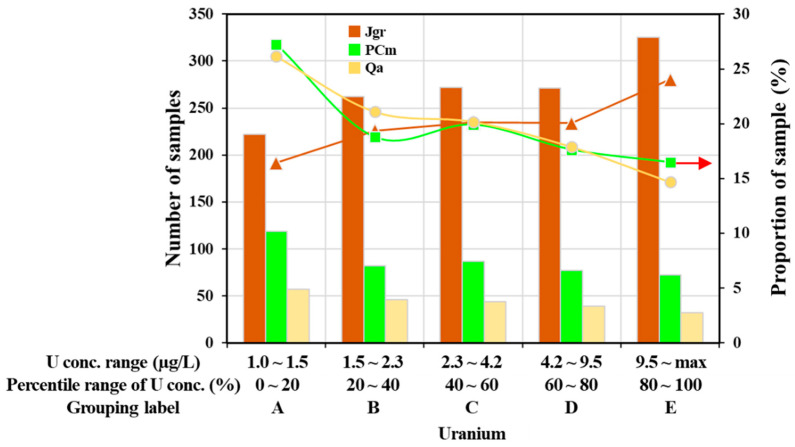
The number and proportion of samples belonging to each group categorized based on the range of uranium concentrations as given in Table 5. The results are correspondent to only three types of geology showing relatively higher excess rates, such as the Jurassic granite (Jgr), the Precambrian metamorphic rock (PCm), and the Quaternary alluvium (Qa). Data from Table 2 and Table 5. The red arrow designates the value of the *y*-axis on the right side.

**Figure 4 toxics-10-00543-f004:**
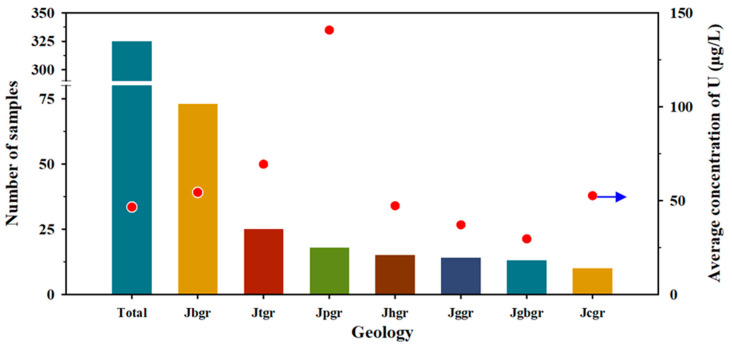
The number of samples and the average concentrations of uranium for the E group according to the detailed rock types of the Jurassic granite (Jgr) showing the highest excess rates, such as the Jurassic biotite granite (Jbgr), the Jurassic two-mica granite (Jtgr), the Jurassic porphyritic granite (Jpgr), the Jurassic hornblende–biotite granite (Jhgr), the Jurassic gneissoid hornblende–biotite granite (Jggr), the Jurassic garnet-bearing granite (Jgbgr), and the Jurassic coarse-grained granite (Jcgr). Data from Table 6. The red dots and blue arrow correspond to the value of the *y*-axis on the right side.

**Figure 5 toxics-10-00543-f005:**
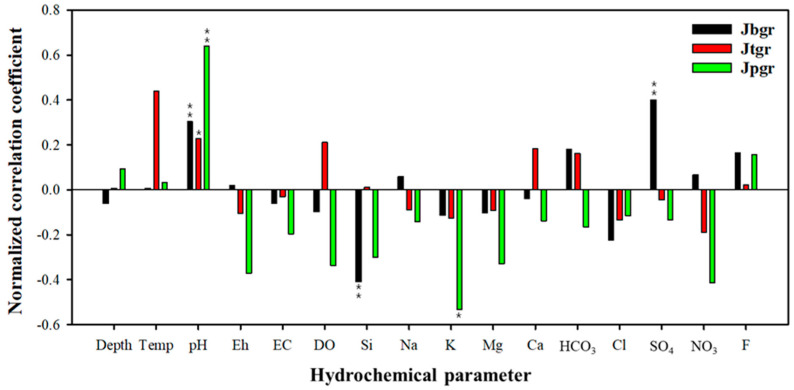
The results of the normalized correlation analyses for three detailed rock types of the Jurassic granite (Jgr), such as the Jurassic biotite granite (Jbgr), the Jurassic two-mica granite (Jtgr), and the Jurassic porphyritic granite (Jpgr), to evaluate the relationship between uranium concentration and hydrochemical parameters. Data from Table 6. * Significance (*p*-value) ≤ 0.05, ** Significance (*p*-value) ≤ 0.01.

**Table 1 toxics-10-00543-t001:** Results of field measurements in groundwater wells for each geology.

Geology		Depth (m)	Temperature (°C)	pH	Eh (mV)	EC (µS/cm)	DO (mg/L)
Total (N = 7036)	Minimum	3.0	6.7	4.5	−300	5.2	0.0
	Maximum	326	20.9	9.4	790	6319	9.0
	Average	69.4	16.3	6.8	220	255	5.1
	Median	70.0	16.1	6.7	203	216	5.5
	Standard deviation	42.5	1.9	0.7	113	176	2.4
Precambrian metamorphic rock (PCm) (N = 1865)	Minimum	8.0	7.1	4.9	−241	6.3	0.0
	Maximum	300	20.9	9.4	767	6319	9.0
	Average	71.4	16.1	6.9	220	232	4.9
	Median	70.0	16.0	6.8	202	190	5.4
	Standard deviation	42.1	1.9	0.7	119	203	2.5
Precambrian sedimentary rock (PCs) (N = 53)	Minimum	10.0	13.4	5.9	125	78.0	0.2
	Maximum	150	19.2	8.3	595	3477	8.9
	Average	66.1	16.2	6.9	317	344	3.4
	Median	60.0	15.9	7.0	318	250	3.0
	Standard deviation	33.2	1.3	0.6	104	468	2.7
Paleozoic sedimentary rock (Ps) (N = 99)	Minimum	10.0	10.7	5.0	−27.6	33.8	0.2
	Maximum	200	20.5	8.5	680	708	9.0
	Average	89.0	15.0	7.4	215	351	5.6
	Median	100	14.6	7.6	227	360	6.4
	Standard deviation	32.8	2.1	0.6	115	144	2.4
Mesozoic igneous rock (Mi) (N = 386)	Minimum	6.0	12.0	5.6	−110	6.8	0.1
	Maximum	300	20.9	9.4	724	1042	8.8
	Average	83.8	17.0	7.0	261	214	5.1
	Median	80.0	17.1	6.9	244	190	5.4
	Standard deviation	41.8	1.8	0.6	128	121	2.1
Mesozoic sedimentary rock (Ms) (N = 391)	Minimum	12.0	11.4	5.5	−150	40.0	0.0
	Maximum	300	20.8	9.4	677	1780	8.7
	Average	83.7	17.4	7.3	250	287	4.6
	Median	80.0	17.3	7.3	251	260	5.2
	Standard deviation	40.3	1.7	0.5	139	173	2.2
Jurassic granite (Jgr) (N = 3136)	Minimum	5.0	6.7	4.5	−300	5.2	0.1
	Maximum	320	20.9	9.0	790	1932	9.0
	Average	63.6	16.0	6.5	217	263	5.5
	Median	60.0	15.9	6.4	201	223	5.9
	Standard deviation	41.3	1.8	0.6	97.3	163	2.3
Cretaceous granite (Kgr) (N = 223)	Minimum	10.0	10.5	5.5	−50.0	26.2	0.1
	Maximum	326	20.9	9.1	642	638	8.8
	Average	86.9	17.1	7.1	222	204	5.2
	Median	90.0	17.2	7.1	199	173	5.5
	Standard deviation	49.0	2.2	0.7	119	117	2.0
Cenozoic igneous rock (Ci) (N = 4)	Minimum	100	14.9	6.2	174	50.0	5.0
	Maximum	150	19.3	7.3	240	310	6.4
	Average	125	16.9	6.8	221	195	5.6
	Median	125	16.6	6.8	236	210	5.6
	Standard deviation	28.9	1.9	0.5	31.7	130	0.6
Quaternary alluvium (Qa) (N = 654)	Minimum	6.0	8.5	5.0	−200	8.1	0.0
	Maximum	300	20.9	8.9	701	1324	9.0
	Average	65.9	16.8	6.7	207	287	4.7
	Median	60.0	16.6	6.6	195	253	5.0
	Standard deviation	44.1	1.7	0.7	116	155	2.5
Okcheon belt (Og) (N = 99)	Minimum	10.0	12.9	5.8	−183	49.5	0.1
	Maximum	200	20.8	9.2	640	639	9.0
	Average	54.7	16.2	7.1	183	214	4.8
	Median	33.0	16.0	7.0	185	195	5.4
	Standard deviation	38.9	1.8	0.7	122	105	2.6
Era unknown (etc) (N = 136)	Minimum	3.0	10.5	5.4	−96.9	34.3	0.1
	Maximum	200	20.7	8.4	676	849	8.9
	Average	73.0	16.9	7.1	153	265	4.7
	Median	80.0	16.8	7.1	118	224	4.8
	Standard deviation	42.2	1.9	0.6	124	145	2.1

**Table 2 toxics-10-00543-t002:** Analytical results of major and minor constituents in groundwater samples for each geology.

Geology		Si	Na	K	Mg	Ca	F	Cl	SO_4_	NO_3_	HCO_3_
		(mg/L)									
Total (N = 7036)	Minimum	0.7	0.7	ND	ND	2.3	ND	ND	ND	ND	5.5
	Maximum	90.4	230	26.0	48.1	249	10.2	248	175	670	390
	Average	11.3	14.4	1.6	5.4	27.6	0.3	17.5	13.0	23.0	84.0
	Median	11.0	10.9	1.1	4.1	22.6	0.0	9.8	9.2	13.2	68.6
	Standard deviation	5.1	13.4	2.0	4.7	20.0	0.9	24.1	14.2	38.8	56.8
PCm (N = 1865)	Minimum	0.7	1.9	ND	0.5	4.0	ND	0.7	ND	ND	5.5
	Maximum	19.7	99.6	26.0	38.4	105	8.4	194	60.0	458	311
	Average	9.9	13.2	1.6	5.0	21.4	0.3	13.2	11.3	17.0	75.7
	Median	9.9	9.3	1.2	3.8	18.7	0.0	7.4	9.0	8.8	61.5
	Standard deviation	3.4	13.9	2.0	4.3	14.0	1.0	17.5	9.2	31.4	51.6
PCs (N = 53)	Minimum	11.9	11.4	1.2	5.4	16.6	ND	4.0	6.0	ND	95.3
	Maximum	27.1	26.1	3.6	20.4	50.9	0.2	20.6	64.0	136	224
	Average	19.5	15.7	1.7	11.1	36.4	0.0	11.3	23.0	30.3	134
	Median	19.0	14.1	1.4	9.3	36.9	0.0	10.2	17.0	13.0	118
	Standard deviation	5.0	5.3	0.9	5.5	11.7	0.1	6.8	21.3	52.2	47.8
Ps (N = 99)	Minimum	5.3	3.5	0.2	ND	5.3	ND	0.8	2.0	ND	36.3
	Maximum	17.4	50.7	3.8	17.7	104	5.5	174	34.0	43.2	237
	Average	8.9	13.9	1.4	6.7	27.5	0.5	23.2	13.0	13.1	94.7
	Median	7.9	8.5	1.0	4.2	17.9	0.2	6.0	9.9	6.5	78.7
	Standard deviation	3.5	11.4	1.1	5.5	23.2	1.2	48.8	8.3	14.1	51.2
Mi (N = 386)	Minimum	1.9	0.7	0.2	0.8	3.5	ND	1.8	ND	ND	6.1
	Maximum	14.6	14.5	7.7	27.8	112	1.0	20.9	38.7	48.4	317
	Average	5.3	3.3	1.4	9.6	42.4	0.1	6.5	16.0	16.0	154
	Median	4.3	2.5	1.0	6.6	42.5	0.0	5.1	13.6	12.2	161
	Standard deviation	3.1	2.8	1.4	7.5	23.0	0.2	4.2	11.4	14.4	85.1
Ms (N = 391)	Minimum	5.3	4.7	ND	0.5	7.7	ND	2.3	0.1	0.1	19.8
	Maximum	90.4	71.1	6.4	15.3	66.2	2.6	47.0	25.7	36.9	215
	Average	15.3	18.3	1.2	5.2	30.0	0.3	16.1	11.6	11.1	91.8
	Median	10.9	13.6	0.9	4.8	29.5	0.1	11.0	10.7	9.4	87.7
	Standard deviation	18.7	15.3	1.2	3.5	14.0	0.6	13.2	6.7	8.4	44.4
Jgr (N = 3136)	Minimum	3.4	3.1	ND	2.1	10.0	ND	2.5	2.8	ND	23.8
	Maximum	19.9	74.2	2.4	34.9	203	1.6	85.3	175	71.7	378
	Average	9.0	17.8	1.0	8.2	45.6	0.2	17.1	27.2	15.9	150
	Median	9.3	11.7	0.9	5.0	34.1	0.0	9.3	13.5	8.8	133
	Standard deviation	3.2	18.0	0.6	7.2	36.8	0.3	18.6	34.4	17.8	98.1
Kgr (N = 223)	Minimum	1.2	2.0	ND	0.1	2.3	ND	ND	ND	ND	9.2
	Maximum	24.4	230	24.1	48.1	249	10.2	248	159	670	390
	Average	11.8	15.1	1.7	5.0	27.9	0.3	19.2	13.1	27.1	78.0
	Median	11.8	11.7	1.1	3.8	22.9	0.0	10.9	9.2	16.5	64.1
	Standard deviation	4.3	13.9	2.2	4.5	20.5	0.9	25.5	15.0	43.2	52.6
Ci (N = 4)	Minimum	4.7	3.5	0.4	0.5	4.8	0.0	1.9	0.0	0.7	21.7
	Maximum	21.2	36.6	1.6	27.6	83.2	2.5	40.2	47.0	34.2	366
	Average	14.0	14.1	0.9	7.0	28.9	0.4	12.5	14.3	12.2	105.4
	Median	14.1	12.7	0.8	5.7	28.3	0.0	8.0	7.0	10.4	81.6
	Standard deviation	5.1	8.5	0.4	7.1	21.9	0.6	11.3	13.9	9.8	83.2
Qa (N = 654)	Minimum	1.7	1.6	ND	0.4	3.0	ND	1.9	ND	ND	13.7
	Maximum	22.0	57.2	11.3	15.0	91.9	9.4	142	43.0	520	216
	Average	12.3	14.8	1.5	5.3	26.3	0.3	18.9	11.3	22.1	85.8
	Median	12.2	12.8	1.1	5.2	23.4	0.0	13.1	9.5	13.4	78.9
	Standard deviation	4.0	9.3	1.7	3.0	15.3	0.9	20.3	8.3	47.3	42.9
Og (N = 99)	Minimum	4.0	1.3	ND	1.4	4.4	ND	4.0	0.6	ND	14.0
	Maximum	12.6	18.9	5.2	27.6	83.7	3.4	86.6	92.6	29.5	240
	Average	8.4	6.1	1.4	5.1	26.9	0.2	11.8	14.6	11.1	87.3
	Median	8.3	4.8	1.0	3.7	23.4	0.0	6.9	7.5	9.6	79.5
	Standard deviation	2.0	3.9	1.2	5.0	18.9	0.6	15.2	19.0	7.6	52.5
etc (N = 136)	Minimum	3.3	6.0	0.5	1.7	9.9	ND	1.6	ND	ND	31.1
	Maximum	22.8	63.4	9.5	42.2	109	0.9	221	47.0	136	316
	Average	14.1	17.3	1.8	7.7	36.7	0.1	23.8	12.9	28.9	109
	Median	14.2	14.8	1.2	6.1	32.3	0.0	10.2	10.0	16.9	98.7
	Standard deviation	3.8	10.4	1.6	6.6	21.6	0.2	41.7	12.0	31.0	58.8

Abbreviations of each geology are identical to those given in Table 1; ND: not detected.

**Table 3 toxics-10-00543-t003:** The hydrochemical facies of groundwater for each geology.

Geology	Number of Wells	Ca-HCO_3_		Ca-Cl		Ca-NO_3_		Ca-SO_4_		Na-HCO_3_		Na-Cl		Na-NO_3_		Na-SO_4_		Mg-HCO_3_		K-NO_3_	
		N	%	N	%	N	%	N	%	N	%	N	%	N	%	N	%	N	%	N	%
Total	7036	5231	74.3	134	1.9	494	7.0	31	0.4	900	12.8	58	0.8	175	2.5	4	0.1	7	0.1	2	0.1
PCm	1865	1472	78.9	36	1.9	68	3.7	2	0.1	249	13.4	11	0.6	23	1.2	0	0	4	0.2	0	0
PCs	53	34	64.2	6	11.3	5	9.4	7	13.2	1	1.9	0	0	0	0	0	0	0	0	0	0
Ps	99	90	90.9	0	0	2	2	2	2	4	4	1	1	0	0	0	0	0	0	0	0
Mi	386	318	82.4	4	1	11	2.9	3	0.8	47	12.2	1	0.3	2	0.5	0	0	0	0	0	0
Ms	391	338	86.5	2	0.5	6	1.5	10	2.6	32	8.2	1	0.3	1	0.3	1	0.3	0	0	0	0
Jgr	3136	2158	68.8	71	2.3	352	11.2	4	0.1	389	12.4	32	1	124	3.9	3	0.1	1	0.1	2	0.1
Kgr	223	171	76.7	0	0	9	4	1	0.5	40	17.9	1	0.5	1	0.5	0	0	0	0	0	0
Ci	4	1	25	0	0	0	0	0	0	3	75	0	0	0	0	0	0	0	0	0	0
Qa	654	467	71.4	11	1.7	32	4.9	1	0.2	106	16.2	11	1.7	24	3.7	0	0	2	0.3	0	0
Og	89	81	91	0	0	1	1.1	1	1.1	6	6.7	0	0	0	0	0	0	0	0	0	0
etc	136	101	74.3	4	2.9	8	5.9	0	0	23	16.9	0	0	0	0	0	0	0	0	0	0

Abbreviations of each geology are identical to those given in Table 1.

**Table 4 toxics-10-00543-t004:** Statistics of uranium concentrations of groundwater for each geology.

Geology	Number of Wells	Number of Wells Exceeding the Uranium Standard	Excess Rate (%)	Uranium Concentration (μg/L)				
				Minimum	Maximum	Average	Median	Standard Deviation
Total	7036	148	2.1	ND	1450	0.4	4.0	28.3
PCm	1865	24	1.3	ND	909	0.2	2.8	25.6
PCs	53	0	0.0	ND	8.6	0.2	0.7	1.6
Ps	99	0	0.0	ND	21.5	0.4	1.0	2.3
Mi	386	2	0.5	ND	283	0.2	1.8	14.6
Ms	391	1	0.3	ND	60.4	0.2	1.1	4.0
Jgr	3136	107	3.4	ND	1450	0.7	6.0	36.5
Kgr	223	3	1.3	ND	95.9	0.4	2.3	7.8
Ci	4	0	0.0	ND	6.8	2.7	3.1	3.5
Qa	654	10	1.5	ND	140	0.4	2.8	11.9
Og	89	1	1.1	ND	37.5	0.2	1.1	4.2
etc	136	0	0.0	ND	14.6	0.6	1.3	2.3

Abbreviations of each geology are identical to those given in Table 1; ND: not detected.

**Table 5 toxics-10-00543-t005:** Grouping of groundwater based on the level of uranium concentrations.

Group	Classification	Percentile Range of Uranium Concentration (%)	Range of Uranium Concentration (µg/L)	Number of Wells (%)
-	Good quality water	-	<1.0	4707 (66.9%)
A	Slightly contaminated water	0–20	1.0–1.5	465 (6.6%)
B	Moderately contaminated water	20–40	1.5–2.3	444 (6.3%)
C	Injuriously contaminated water	40–60	2.3–4.2	482 (6.9%)
D	Highly contaminated water	60–80	4.2–9.5	471 (6.7%)
E	Severely contaminated water	80–100	9.5–1450	467 (6.4%)

**Table 6 toxics-10-00543-t006:** The uranium concentrations and hydrochemical parameters of five groups for three types of geology showing relatively higher excess rates.

Parameter	Jurassic Granite (Jgr)					Precambrian Metamorphic Rock (PCm)					Alluvial Layer (Qa)				
	A	B	C	D	E	A	B	C	D	E	A	B	C	D	E
Number of wells	221	263	272	271	325	119	82	87	77	72	57	46	44	43	32
Median of uranium concentration (μg/L)	1.2	1.8	3.2	6.1	19.7	1.2	1.8	3.0	5.6	19.0	1.1	1.8	3.1	5.6	22.9
Depth (m)	70	80	85	100	100	80	80	100	100	100	80	75	100	100	100
Temp (°C)	15.8	15.8	15.9	15.9	16.1	15.8	15.9	15.7	16.0	15.1	16.9	16.5	16.1	16.2	16.5
pH	6.5	6.5	6.7	6.7	7.0	6.9	7.1	7.3	7.1	7.5	6.6	6.7	6.8	7.0	7.2
Eh (mV)	201	190	183	179	168	191	181	162	220	158	190	187	176	154	163
EC (µS/cm)	203	209	221	233	252	217	237	230	204	198	280	214	279	286	341
DO (mg/L)	6.1	6.0	5.7	5.3	5.0	5.1	4.4	4.9	4.4	4.2	4.8	5.0	4.5	4.6	4.9
Si (mg/L)	12.5	13.3	12.4	12.8	11.4	10.1	9.9	9.2	9.9	8.8	11.7	12.4	13.4	13.3	12.7
Na (mg/L)	10.6	12.0	12.3	11.9	12.9	10.3	9.7	10.5	10.8	9.9	14.2	14.2	13.6	14.0	17.1
K (mg/L)	1.0	1.0	1.1	1.0	0.9	1.1	1.2	1.0	1.1	0.8	1.4	1.2	1.1	1.3	1.3
Mg (mg/L)	3.7	3.5	3.7	3.3	3.3	4.8	5.9	4.4	5.2	3.3	5.0	4.3	5.9	5.9	4.8
Ca (mg/L)	21.7	22.5	24.4	26.4	30.6	22.4	27.6	25.2	24.5	23.1	27.1	21.7	32.7	31.6	38.4
HCO_3_ (mg/L)	65.9	72.5	74.6	83.4	96.4	85.4	106	105	102	86.5	83.9	79.4	109	103	104
Cl (mg/L)	7.9	9.6	10.3	8.3	10.0	7.1	6.8	6.5	6.3	3.7	14.6	12.6	13.5	16.6	12.4
SO_4_ (mg/L)	5.8	6.0	7.0	6.9	9.0	9.0	8.1	10.0	7.1	8.0	8.5	8.0	9.4	13.0	7.7
NO_3_ (mg/L)	12.5	15.0	14.6	11.2	10.5	8.0	5.6	4.9	5.1	4.2	17.0	10.8	12.5	10.2	10.3
F (mg/L)	ND	ND	ND	ND	ND	ND	ND	0.1	0.1	0.3	ND	ND	ND	0.2	0.3

ND: not detected.

**Table 7 toxics-10-00543-t007:** Normalized correlation coefficients between uranium concentration and hydrochemical parameters for three types of geology showing relatively higher excess rates.

Parameter	Total	Jurassic Granite (Jgr)					Precambrian Metamorphic Rock (PCm)					Alluvial Layer (Qa)				
		A	B	C	D	E	A	B	C	D	E	A	B	C	D	E
Depth	0.34	0.03	−0.12	−0.15	−0.07	0.11	−0.23 *	0.37 **	0.29 **	0.45 **	0.31 **	0.15	0.28 **	0.08	−0.14	−0.21 *
Temp	0.06 *	−0.31 **	0.19 *	0.32 **	0.18 *	−0.02	0.24 **	−0.33 **	0.47 **	−0.04	−0.02	0.09	0.37 **	−0.05	0.07	−0.18 *
pH	0.23	0.04	0.16	0.05	−0.03	0.23 *	−0.03	−0.06	0.23 *	−0.00	0.13	0.00	0.31 **	0.21 *	0.15	−0.28 **
Eh	−0.06	0.16	−0.17	0.06	0.16	−0.33 **	0.23 *	0.01	−0.33 *	−0.05	−0.13	0.18	−0.15	−0.16	−0.28 **	−0.12
EC	0.02	−0.01	−0.18 *	0.04	0.10	−0.14	−0.00	0.12	−0.10	−0.15	−0.10	0.11	−0.03	−0.11	0.15	0.02
DO	0.09	−0.26 **	−0.00	0.19 *	−0.09	−0.22 *	0.09	0.22	−0.18	−0.18	0.19 *	0.37 **	−0.01	0.46 **	−0.14	0.34 **
Si	−0.10	0.13	0.33 **	0.08	−0.08	−0.17	−0.07	0.02	0.02	−0.10	−0.01	0.18	−0.10	0.06	−0.34 **	−0.19 *
Na	0.07	−0.03	−0.07	0.18 *	−0.05	−0.12	−0.01	−0.17	−0.04	−0.07	0.13	0.04	−0.20 *	−0.42 **	−0.03	0.16
K	−0.16	−0.22 *	−0.17	0.17	0.07	−0.02	0.09	0.06	−0.28 **	0.03	−0.21 *	−0.04	−0.16	−0.07	−0.40 **	0.33 **
Mg	−0.03	0.08	−0.14	−0.03	0.05	−0.08	0.03	0.24 **	−0.09	−0.12	−0.11	0.07	0.01	−0.08	0.03	0.00
Ca	0.00	0.04	−0.17	−0.04	0.07	−0.10	0.00	0.19 *	−0.09	−0.12	−0.17	−0.05	0.13	0.11	0.24 **	−0.04
HCO_3_	0.00	0.18	0.01	−0.03	0.07	−0.05	0.13	0.30 **	−0.14	0.08	−0.12	−0.20 *	0.16	−0.14	0.43 **	−0.01
Cl	0.06	0.00	−0.12	0.04	−0.03	−0.15	0.00	0.03	0.12	−0.24 **	−0.03	0.28 **	−0.11	−0.05	−0.20 *	−0.18 *
SO_4_	0.01	−0.12	−0.29 **	0.05	−0.10	0.06	0.02	0.06	−0.11	0.04	−0.18 *	−0.26 **	0.06	0.07	0.12	0.41 **
NO_3_	−0.17	−0.24 **	−0.25 **	0.16	0.12	−0.15	−0.24 **	−0.26 **	−0.25 **	0.10	−0.03	0.30 **	−0.20 *	−0.07	−0.24 **	0.18 *
F	0.12	−0.06	−0.05	0.25 **	0.14	0.07	−0.18 *	−0.23 *	0.24 **	0.19 *	0.46 **	0.04	−0.05	−0.26 *	0.35 **	0.12

* Significance (*p*-value) ≤ 0.05, ** Significance (*p*-value) ≤ 0.01.

## Data Availability

The data presented in this study are available on request from the corresponding author.

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
