# Peer review of "Uranium Concentrations in Private Wells of Potable Groundwater, Korea"

_toxics, 2022, doi:10.3390/toxics10090543_

Round 1

Reviewer 1 Report

Dear authors,

Here is my review of the Manuscript ID: toxics-1922859, entitled “Uranium concentrations in private wells of potable groundwater, Korea” by Woo-Chun Lee, et al. This is a very interesting study of more than 7 000 groundwater samples from private wells in Korea showing the existence of numerous of wells with U concentrations exceeding the Korean and WHO standard (30 mg/L). Authors assessed the potential health risk. In my opinion this paper can be published in the Toxics journal, after a minor revision.

Specific comments

Format of the Tables 2, 3, 6, 7.   They are not well readable, they should be improved.

The average values of so big number of samples could be rounded off (text & Tables), for example at the line 145, Eh (317 mV) instead Eh (316.9 mV).

Table 2. Please use a symbol for values lower than the detection of the analytical method used, instead 0.0

Tables 2 & Table 3. Please check again the order and correct.

Please check again and correct, for example, Table 5, Figure 3….Percentile ?

Figures 2, 3, 4, 5: Please add the data source. Table….ref.  ?

Best wishes

Author Response

Please refer to the attached file titled: "(Response to Reviewer 1) toxics-1922859".

Reviewer 2 Report

Woo Chun Lee et al. tried to solve the big problem of groundwater uranium pollution in private wells of potable groundwater in Korea. Such studies of U and Sr with a social or natural science bias have been carried out recently, for example, in the USA (see paper https://doi.org/10.1016/S2542-5196(22)00043-2 and https://doi.org/10.1016/j.apgeochem.2020.104867). However, compared to these articles, in the study of Woo-Chun Lee et al. social or natural science components are not visible. It is not clear when and how field studies were carried out, for what purpose the physicochemical characteristics of groundwater were determined, and what kind of correlations between uranium concentrations and these parameters the authors expected to obtain. As a result, a number of cumbersome very generalized statistical tables and the only trivial conclusion about the relationship between elevated uranium concentrations and chemical weathering of granites were obtained.

Maybe a well-thought-out graphical display of the results obtained on maps and graphs would reveal implicit patterns and arouse the interest of readers (see examples at the above links).

Minor

Page 1, Line 23

Instead of the WHO standard (30 mg/L), it should be 0.030 mg.L-1 (chemical aspect). In addition, the guidance level of 234U is 1 Bq.L-1 (radiological aspect) and screening levels for drinking-water, below which no further action is required, are 0.5 Bq/l for gross alpha activity (https://doi.org/10.1016/j.jenvrad.2015.11.006)

Page 2, Lines 46-48

The regulations of the World Health Organization (WHO) and several advanced countries also state that the guidance level of 234U is 1 Bq.L-1 (radiological aspect) and screening levels for drinking-water, below which no further action is required, are 0.5 Bq.L-1 for gross alpha activity

Page 2, Lines 50-52

Levels in drinking-water are generally less than 1 µg/l, although concentrations as high as 700 µg/l have been measured in private supplies (WHO, Page 430).

Page 2, Lines 63-65

Have you compared it to India?

Page 2, Lines 92-93

You write correctly:

Based on the geology presented by the 250,000:1 map, the geology and geological period of the well-sitting sites were largely classified into 11 groups (Table 1).

However, on Page 5, Lines 104-106 you are already spelling wrong:

The primary hydrochemical properties of the groundwater were measured in the field, and the major and minor constituents were measured in the laboratory (Tables 1 and 2).

Table 3 is desirable to expand to the full page for better perception.

Instead of the Table 3, it should be Table 2 (The numbering of tables in the manuscript must be consecutive)

Table 2 is desirable to expand to the full page for better perception.

Page 6, Lines 157-158

Your statement "In terms of major anions, bicarbonate showed the highest concentration, and those of the other constituents were SO4 > NO3 > Cl (Table 2)" is not true when it comes to Average Total values: HCO3 (84) > NO3 (23) > Cl (17.5) > SO4 (13). Statements of this kind should be supported by graphic illustrations.

Also not Ca > Na > Mg > K, but Ca > Na > Si > Mg > K

Page 14, Lines 330-331

Instead of the standard (30 mg/L) for uranium levels in drinking water, it should be 0.030 mg.L-1

Page 15, Lines 349-350

See Table 9.4 Treatment performance for some common radionuclides

Author Response

Please, refer to the attached file titled: "(Response to Reviewer 2) toxics-1922859".

Round 2

Reviewer 2 Report

The authors have addressed all my comments to great extent

 I recommend paper publication
